# Acid–Base Properties of Oxides Derived from Oxide Melt Solution Calorimetry

**DOI:** 10.3390/molecules28124623

**Published:** 2023-06-07

**Authors:** Alexandra Navrotsky, Anastasia Koryttseva

**Affiliations:** 1School of Molecular Sciences, Center for Materials of the Universe, Arizona State University, P.O. Box 871604, Tempe, AZ 85287-1604, USA; 2School for Engineering of Matter, Transport and Energy, Center for Materials of the Universe, Arizona State University, P.O. Box 871604, Tempe, AZ 85287-1604, USA; 3School of Earth and Space Exploration, Center for Materials of the Universe, Arizona State University, P.O. Box 871604, Tempe, AZ 85287-1604, USA; 4Department of Chemistry, Lobachevsky State University of Nizhniy Novgorod, Gagarin Avenue 23, 603022 Nizhniy Novgorod, Russia; koak@chem.unn.ru

**Keywords:** oxide melt solution calorimetry, acid–base properties, thermodynamics, electronegativity

## Abstract

The paper analyzes the relationships among acid–base interactions in various oxide systems and their thermodynamics. Extensive data on enthalpies of solution of binary oxides in oxide melts of several compositions, obtained by high-temperature oxide melt solution calorimetry at 700 and 800 °C, are systematized and analyzed. Oxides with low electronegativity, namely the alkali and alkaline earth oxides, which are strong oxide ion donors, show enthalpies of solution that have negative values greater than −100 kJ per mole of oxide ion. Their enthalpies of solution become more negative with decreasing electronegativity in the order Li, Na, K and Mg, Ca, Sr, Ba in both of the commonly used molten oxide calorimetric solvents: sodium molybdate and lead borate. Oxides with high electronegativity, including P_2_O_5_, SiO_2_, GeO_2_, and other acidic oxides, dissolve more exothermically in the less acidic solvent (lead borate). The remaining oxides, with intermediate electronegativity (amphoteric oxides) have enthalpies of solution of between +50 and −100 kJ/mol, with many close to zero. More limited data for the enthalpies of solution of oxides in multicomponent aluminosilicate melts at higher temperature are also discussed. Overall, the ionic model combined with the Lux–Flood description of acid–base reactions provide a consistent and useful interpretation of the data and their application for understanding the thermodynamic stability of ternary oxide systems in solid and liquid states.

## 1. Introduction

“Acid” and “base” were among the first terms to appear at the dawn of chemistry to describe readily observable properties of substances and solutions, e.g., a sour taste, a soapy feel, or corrosive properties. Definitions and concepts evolved as the science progressed [1]. Though originating from aqueous systems, the concept of acidity/basicity is now much more general and applicable to a variety of systems, including solids, molten salts, glasses, magmas, and many others. Nevertheless, consistent extension of acid–base concepts to different chemical systems is difficult, and a universal quantitative acid–base scale may not be attainable. Rather, molecules, ions, and solids show acid–base behavior with respect to each other, defining appropriate equilibria in a given system or set of related systems. These equilibria can be described at the macroscopic scale by thermodynamics, while at the microscopic scale they are determined by quantum mechanics and chemical bonding. Acid–base theories (see below) incorporate both thermodynamic and molecular constraints. Acid–base concepts have been applied fruitfully in oxide systems in the solid, molten, and glassy states, as well as in vaporization processes [2,3,4,5,6,7,8]. It is common in the ceramic, glass, and geochemical fields to call calcia a basic oxide, to call alumina amphoteric, and to call silica acidic, with the realization that the strongest (most exothermic) interactions occur between a highly acidic and highly basic oxide, which form stable and high melting compounds. The present paper explores the acid–base character of oxide systems in three ways. First, we briefly summarize prior concepts of the acid–base character, including those in older literature, which may not be familiar to current readers, and we show how these have already been applied in oxide systems. In the main part of the paper, we document the enthalpies of solution of a wide range of crystalline binary oxides in several different molten oxide systems at different temperatures. These systematic data were obtained via oxide melt solution calorimetry in order to measure the heats of formation of multicomponent oxide ceramics and minerals [9,10,11]. These enthalpy of solution data, taken together, define common trends and a semiquantitative acid–base scale, which are then useful for understanding the thermodynamic driving forces for formation and decomposition reactions. Finally, we present some applications of these concepts to complex aluminosilicate melts.

## 2. Acid–Base Concepts and Definitions

Arrhenius defined an acid as a substance that dissociates with the formation of H^+^ ions and a base as a substance that dissociates with the formation of OH^−^ ions. The neutralization reaction (acid–base interaction) is written as
H^+^ + OH^−^ = H_2_O

Then, independently of each other, Brønsted and Lowry [12,13] considered a proton donor to be an acid, and its acceptor to be a base. This was applied to reactions in an aqueous solution and includes the Arrhenius definition as a special case. The concepts can be generalized and applied to any substances that can exchange protons.

Lux and Flood [14] described the properties of acids and bases based on the behavior of the oxide ion O^2−^. An acid is an oxide ion acceptor and a base is an oxide ion donor. For example, the following reaction can take place among solids or in a high-temperature melt.
SiO_2_ + 2CaO = Ca_2_SiO_4_

SiO_2_ is an O^2−^ ion acceptor, so it is an acid; CaO is an O^2−^ ion donor, so it is a base. The oxide ion is transferred from being associated with Ca in lime to being bound in a silicate species, in this case the orthosilicate anion (SiO_4_)^4−^. The base CaO reacts with the acid SiO_2,_ to form the salt Ca_2_SiO_4_. The Lux–Flood concept is equally applicable to reactions in glasses, melts and crystalline phases.

The Lewis definition [15] was developed from the direction of electron pair transfer. A base is an electron pair donor and an acid is its acceptor. All the above reactions are consistent with the Lewis definition and can also be applied to reactions where molecules rather than ions are formed. Thus, all reactions caused by the transfer of a proton, an oxide ion, the autoionization of a solvent, or the occurrence of reactions with the formation of adducts and complex compounds are acid–base reactions according to the Lewis definition.

The Usanovich definition [16] can be applied to any chemical reaction resulting in salt formation. An acid is a cation donor or an anion acceptor. A base is a substance that can accept a cation or donate an anion during a chemical reaction. Thus, exchange (metathesis) and oxidation–reduction reactions can be considered acid–base reactions. This concept of acids and bases includes substances of various classes: acids in the classical definition, neutral atoms and molecules, salts, and positive and negative ions, united only because they all possess a common function as donors or acceptors. Therefore, the Usanovich concept can be called the theory of generalized acids and bases.

The question of the influence of the medium on acid–base properties becomes a local issue in the theory of generalized acids and bases; a solvent medium is not required at all. Therefore, the Usanovich concept of acids and bases should play an important role in solid-state reactions in crystal chemistry and geochemistry, as well as in the chemistry of liquid solutions including aqueous phases, silicate melts and molten salts.

## 3. Acid–Base Strength from the Viewpoint of Chemical Bonding

Urusov [17] showed that the Usanovich definition of generalized acids and bases is related to chemical bonding through the concept of electronegativity (EN), which relates to the properties of atoms (ions and radicals) and the degree of ionicity (covalence) of the chemical bond. Since EN is the ability of an atom to attract an electron in a molecule, it is approximately proportional to the ionization potential of the valence electron of the atom.

Hardness and softness are concepts of chemical bonding defined as the tendency of a compound to form bonds of a predominantly ionic or covalent nature, respectively. There are empirical and semi-empirical methods for the quantitative characterization of the hardness and softness of generalized acids and bases, which allow one to correlate their strength with electron density, molecular structure, or other factors [18]. Klopman [19] quantified the hardness and softness of acids and bases using the semi-empirical molecular orbital (MO) method. He equated softness with the orbital electronegativity of those valence orbitals that are free in acceptors (acids) and completely occupied in donors (bases). He calculated the hardness parameters not only for free atoms, but also for ions in aqueous solution.

Returning to the solid state, the hardness of an atom or ion in a crystal is well described by its total electronegativity. Taking into account the Madelung effect, the hardness parameter in an ionic crystal is closely related to the Cartledge potential (charge/radius = z/r). Acid–base scales based on spectroscopic data have also been proposed [20] but will not be discussed further here, as our emphasis is on thermodynamics.

## 4. Acid–Base Strength from the Viewpoint of Thermodynamics

Forming a salt from an acid and a base lowers the free energy of the system, and the magnitude of the change in free energy should be a measure of the difference in acid–base properties of the reactants. Thus, thermodynamic data provide direct insight into acid–base reactions.

Ramberg [21] examined the regularities of the heats of formation of complex oxygen-containing compounds depending on the polarizability of the cations forming them. The more different the acid–base character of the two constituents, the more exothermic the formation enthalpy. The heats of formation from oxides of different oxosalts of the same metal become more exothermic in the direction of decreasing polarity of the covalent bond in the oxyanion in the direction B, C, N and Si, P, S, and Cl (i.e., borates, carbonates, nitrates, orthosilicates, phosphates, sulfates, and perchlorates).

Kireev [22] analyzed the periodicity of the thermodynamic properties of oxides and oxygen salts. He used the Gibbs functions of the reactions of formation of these complex compounds from simple oxides, with ∆G_f,ox_^0^ becoming more negative the more the constituent oxides differed in acid–base properties. If one chooses a specific reference oxide, then the remaining oxides can be arranged in a well-defined sequence relative to the selected standard. Kireev chose Na_2_O as a reference for acidic oxides and SO_3_ for basic oxides. Then, he arranged basic oxides (in relation to SO_3_) in order of decreasing ∆G_f,ox_^0^ of the corresponding sulfates, and acid (in relation to Na_2_O) oxides in order of increasing ∆G_f,ox_^0^ of their compounds with Na_2_O.

For a significant number of salts and mixed metal oxides, there are data only on ∆H_f,ox_^0^ of their formation from oxides, but not on ∆S_f,ox_^0^ or ∆G_f,ox_^0^. An analysis of data on compounds for which all thermodynamic parameters are available shows the following. When reactants and products are solids, the entropy of formation term is generally small in magnitude and often can be ignored compared to the very exothermic enthalpy of formation, especially for the interaction of strongly basic and strongly acidic oxides. The effect of the entropy increases for reactions involving gaseous oxides and can be dominant when the enthalpy term is small in magnitude. Metal oxides of a similar type, belonging to the same group of the periodic table, maintain the same sequence in enthalpy or free energy in compounds with different acidic oxides. However, this regularity breaks down for less similar oxides.

In Gutmann’s method of donor numbers (DN) [23], the sequence of Lewis acids also depends on the substance taken as a reference. If one fixes the reference base, then the Lewis acids can be arranged in a series based on the heats of the neutralization between the reference base and the Lewis acid. Gutmann chose SbCl_5_ as a standard substance for donor ability evaluation. Experimentally measuring the enthalpy of formation of only two adducts of a given acid, one can predict the enthalpy of formation of adducts with any other donor solvent for which the DN is known.

These previous authors used thermodynamic data obtained via different methods with different systematic and random errors. Both the paucity of thermodynamic data and the inherent inconsistencies in data obtained by different methods limit the accuracy of such correlations. Thus, an extensive bank of reliable and consistent thermodynamic data would serve as background for a variety of further calculations, comparisons and analyses. The goal of the present work is to use an extensive set of calorimetrically measured enthalpies of solution of simple binary oxides in several molten oxide solvents at high temperature as the basis for defining and correlating their acid–base properties.

## 5. High-Temperature Oxide Melt Solution Calorimetry, Acid–Base Reactions and Properties of Molten Oxide Solvents

This calorimetric methodology has been developed since the 1960s and is now well-established and used to measure enthalpies of formation and transformation in simple and complex oxides and other systems [11,24]. It utilizes a high-temperature calorimeter, initially custom-built but now commercially available as the Setaram Alexsys and MHTC96 instruments, to contain several grams of a molten oxide solvent into which milligram amounts of solid samples are dissolved at high temperature (typically 700 to 1500 °C). The difference in the measured enthalpy of solution of reactants and products directly gives the enthalpy of reaction. Although early experiments positioned a small sample on a platinum holder directly above the solvent in the calorimeter and stirred it into the melt, measuring the heat of solution directly, current protocols favor dropping a pellet of sample from room temperature into the molten solvent. The measured enthalpy, commonly called the enthalpy of drop solution, Δ_ds_H(T), consists of two terms. The first is the enthalpy associated with heating it from room temperature to a calorimetric temperature. The second is the enthalpy of solution. Thus,
$$\Delta_{\mathrm{ds}}H(T)=\int_{298}^{T}C_{p}dT+\Delta_{s}H(T)$$

In some cases, other reactions, such as gas release, oxidation, or reduction, occur in the solvent and must be included in the thermochemical calculations. In general, these are not directly relevant to acid–base reactions, and oxides undergoing such processes of dissolution in the calorimetric solvent have been left out of our dataset.

Two calorimetric solvents, typically used at 700 and 800 °C, have emerged as the most useful ones for a variety of oxide systems: sodium molybdate of the composition 3Na_2_O·4MoO_3_ and lead borate of the composition 2PbO·B_2_O_3_ [10,11,12]. Other alkali borates, molybdates, and tungstates have also been used, but to a smaller extent [25,26,27]. The enthalpies of a drop solution of a smaller set of oxides in molten silicates above 1000 °C have also been measured and are included in the paper as an application. The extensive use of the lead borate and sodium molybdate solvents over many years has generated a large set of drop solution enthalpies, which are summarized in Table 1. Because of the low concentration of dissolved oxide, these enthalpies refer to Henry’s law, where the enthalpy of the drop solution does not depend on the amount of solute dissolved or the presence of small amounts of other solutes in the melt. The measured enthalpies of drop solution of binary oxides have been consistent over many years and among different laboratories, with typical uncertainties of ±1–2%. Thus, these calorimetric data offer a unique opportunity for the thermodynamic analysis of the acid–base character of solid oxides and contribute to the main goal of this paper.

Before proceeding to that discussion, some comments on the acid–base chemistry of the solvent systems themselves are in order. High-temperature calorimetry has been used to directly measure enthalpies of mixing in a number of oxide glasses and melts, giving insight into their acid–base equilibria. The systems studied include PbO–V_2_O_5_ [41], PbO–B_2_O_3_ [42], PbO–SiO_2_ [43], PbO–GeO_2_ [44], Na_2_O–MoO_3_ [45], Li_2_O–B_2_O_3_, Na_2_O–B_2_O_3_ and K_2_O–B_2_O_3_ [27]. In some cases, integral enthalpies of mixing were measured by combining large amounts of components; in others, partial molar enthalpies of solution of one or both of the components in binary systems were measured as a function of melt composition. The selected data are summarized in Figure 1.

In the system Na_2_O–MoO_3_, Na_2_O is a strongly basic oxide, while MoO_3_ is a strong acid. We can write a number of successive stages of acid–base interaction that occur when MoO_3_ is added to a melt initially rich in Na_2_O. The species in the melt, with increasing MoO_3_ content, are MoO_4_^2−^, Mo_2_O_7_^2−^ and MoO_3_. Their equilibria are described by the following:O^2−^(in melt) + MoO_3_ (s) = MoO_4_^2−^ (in melt)
MoO_4_^2−^ (in melt) + MoO_3_ (s) = Mo_2_O_7_^2−^ (in melt) 
MoO_3_ (s) = MoO_3_ (in melt)

When the melt contains a mixture of several types of molybdate species, such as those above or their polymers, it is a buffered acid–base system (Figure 1a). There are several buffer regions in the Na_2_O–MoO_3_ system [45]. For 0.52 < n (MoO_3_) < 0.63 and n (MoO_3_) > 0.7, where the enthalpy of solution of MoO_3_ changes very slowly with increasing mole fraction of MoO_3_, these buffer regions contain mixtures of molybdate species. The composition chosen as a calorimetric solvent (3Na_2_O 4MoO_3_) is a buffer containing mainly MoO_4_^2−^ and Mo_2_O_7_^2−^ species charge-balanced by Na^+^ ions.

The data for the PbO–B_2_O_3_ system (Figure 1b) can be interpreted similarly [42]. In this system, PbO is a moderately strong base (Pb^2+^ is a weak acid), while B_2_O_3_ is a strong acid. The interaction between B_2_O_3_ and PbO represents a strong acid–base reaction:(1 − n) PbO + n B_2_O_3_ → (1 − n) Pb^2+^ + 2n {boron-based anion(s)}

In a PbO–B_2_O_3_ melt, there is a buffer region in the composition range 0.25 < n (B_2_O_3_) < 0.5, where partial molar enthalpies of solution change only to a moderate degree and the melt acidity changes little with the change in composition. Because the borate anions tend to polymerize, their types and amounts depend on (B_2_O_3_). Such speciation has been analyzed in detail [24]. The composition used as a calorimetric solvent, 2PbO·B_2_O_3_ with n (B_2_O_3_) = 0.33), lies within this buffer region. The dependence of the enthalpy of solution on the mole fraction of acidic oxide is similar in appearance to that of an aqueous acid–base titration curve. Buffer regions are separated by rapidly changing regions of enthalpy, which is very clearly seen in Figure 1c,d. The relative strength of acids and bases can be judged by the magnitude of the enthalpy “jump” on the curve (the distance between the two relatively flat lines defining buffer regions). For example, Figure 1d shows that the enthalpy jump increases in the order Li_2_O–B_2_O_3_, Na_2_O–B_2_O_3_, K_2_O–B_2_O_3_, indicating an increase in basic properties in this direction, and is consistent with the general trends in properties in the periodic table.

## 6. Enthalpies of Solution of Crystalline Binary Oxides in Molten Oxide Solvents as a Probe of Their Acid–Base Character

Table 1 summarizes the enthalpies of drop solution and of solution of crystalline binary oxides in lead borate and sodium molybdate solvent at 700 and 800 °C. Table 2 summarizes some data for solution of oxides in lithium sodium borate melt at 800 °C and in aluminosilicate melts at 1450 °C. The data are shown per mole of oxide ion because that is the species directly interacting by acid–base reactions with the complex anionic species in the melt.

Following the discussion above, which suggests close relationships between hardness/softness, electronegativity, and acid–base character, we plot the enthalpy of solution vs. the Pauling electronegativity [1] (Figure 2 and Figure 3). Though the data show some scattering, they adhere to well-defined trends. First, the dependence of enthalpy of solution on electronegativity is not monotonic. More exothermic enthalpies of solution are seen for oxides with low and high electronegativity than for those in the intermediate range. Taken individually for specific systems (Figure 2) or together for all systems (Figure 3), the heat of solution data show roughly parabolic trends with electronegativity. the dependence is adequately described by a quadratic polynomial. We stress that this polynomial (or another one of a higher order to better fit the data) has no physical significance; it is an aid to the eye and may offer a first estimate for the enthalpy of solution of oxides whose enthalpies of drop solution have not yet been measured or cannot be measured, for example because of volatility.

From a somewhat different vantage point, the dependence of the enthalpy of solution on electronegativity can be separated into three groups. Oxides with low electronegativity (less than about 1.3), which are strong oxide ion donors and therefore strong bases, show enthalpies of solution of more negative than −100 kJ/mol. These include the alkali and alkaline earth oxides, for which the enthalpy of solution becomes more negative with decreasing electronegativity in the order Li, Na, K and Mg, Ca, Sr, Ba in both solvents (Figure 3). The values in sodium molybdate are somewhat more negative than those in lead borate, which is consistent with the former being a somewhat more acidic solvent.

The second group (Figure 3) is oxides with high electronegativity (>2.1), including P_2_O_5_, SiO_2_, GeO_2_ and other acidic oxides. These oxide ion acceptors (acids) clearly compete with the complex anions in the solvent for bonding to oxygen. They dissolve more exothermically in the less acidic solvent (lead borate). The remaining oxides, a large group with electronegativity between 1.3 and 2.1 (Figure 3), have heats of solution between +50 and −100 kJ/mol with many close to zero. They correspond to oxides often referred to as amphoteric.

When choosing electronegativity, we decided to use the Pauling scale, because the electronegativities of almost all elements are given. Moreover, all EN scales are in good agreement with each other. For comparison, we plot the enthalpy of solution from the Mulliken EN of elements [51] (Figure 4), and obtain similar results as those obtained when using the Pauling scale:

The EN concept applies not only to elements, but also to ions and compounds, including oxides. There are different approaches to the calculation of EN of compounds: the arithmetic mean, $EN\left(M_{x}O_{y}\right)=\frac{x\cdot EN\left(M\right)+y\cdot EN\left(O\right)}{x+y}$ [52], and geometric mean,$\;EN\left(M_{x}O_{y}\right)={\left(EN{\left(M\right)}^{x}\cdot EN{\left(O\right)}^{y}\right)}^{1/\left(x+y\right)}\;$ [53] from the consistent elements. Plots of the enthalpy of solution vs. these EN parameters give similar trends as those calculated using the EN of elements (Figure 5a,b).

For detailed studies, it is useful to classify oxides and analyze their properties in smaller groups based on the location of the oxide-forming element in the periodic table, reflecting the type of valence electrons and structural types, e.g., fluorite, rutile, and rocksalt. Since stoichiometry and the ionic radius are the main determinants of the structure and properties in ionic systems, we plot the enthalpies of solution of oxides versus the ionic radius of the corresponding element (the radius corresponding to the coordination environment and oxidation state of the element in the structure of the corresponding oxide [54]), dividing them into six stoichiometric groups: M_2_O, MO, M_2_O_3_, MO_2_, M_2_O_5_, and MO_3_. For M_2_O (alkalis), MO (alkaline earth and d-block elements), and M_2_O_3_ (oxides of aluminum, manganese, iron, gallium, and lanthanides), the enthalpies of solution in all melts become more exothermic with an increase in ionic radius. The dependence is well-fitted with a second-order polynomic (Figure 6a–c), confirming a stronger-than-linear dependence of enthalpy on ionic size. The group of MO stoichiometry is dominated by divalent elements that form oxides of a rocksalt structure. The M_2_O_3_ stoichiometry group includes oxides of the C-type lanthanides. The similar behavior for all three groups corresponds to the strengthening of the basic properties in a given group with increasing number in the periodic table. For oxides of M_2_O_5_ and MO_3_ stoichiometry (Figure 6e,f) with acidic properties, the exothermicity decreases with an increasing radius (decreasing electronegativity), reaches zero, and then the enthalpy of solution becomes slightly positive. For oxides of MO_2_ stoichiometry, the dependence does not show a clearly defined trend, which is consistent with the amphoteric nature of these oxides (Figure 6d). The consistency of the results for different systems confirms the good applicability of the ionic model for this type of interaction.

## 7. Application in Complex Aluminosilicate Melts

It is obvious that a systematic tabulation of the enthalpies of solution of oxides in different molten oxide solvents (Table 1) is useful for further calorimetric studies of the thermodynamics of the formation of complex oxides. The current emphasis on new materials for energy applications, including batteries and fuel cells, increasingly targets multicomponent systems, both in terms of minor doping to control properties and in terms of systems containing comparable amounts of several oxides, sometimes called high-entropy materials. Many groups worldwide are now performing oxide melt solution calorimetry and a consistent reference set of enthalpies of solution of binary oxides should be used in such studies. The data in the present paper update a set of values published in 2014 [11]. Linking these values to acid–base properties provides atomistic understanding, may help identify uncertain values in individual experiments, help in choosing appropriate molten oxide solvents for calorimetry and for crystal growth, and help develop new solvents.

From the vantage points of Earth and planetary science, ceramics science and materials science, the dissolution of oxides in silicate melts is of major importance. Natural silicate melts (magmas in planetary interiors and lavas on their surfaces) are largely molten aluminosilicates at temperatures between 900 and 1500 °C. The parent melts, with major constituents in the CMAS (calcia–magnesia–alumina–silica) system, also contain significant amounts of iron oxide and titanium oxide and minor trace amounts of all other elements. Processes of solution, crystallization, phase separation and equilibration with H_2_O and CO_2_ lead to a rich variety of rocks formed in metamorphic and igneous environments. The minerals formed and their compositions are largely governed by the thermodynamics of crystal–melt equilibria, and are thus controlled by the acid–base properties of these melts. In ceramic systems, “fluxes”, i.e., low melting oxide or fluoride mixtures, are used to grow crystals of technologically important compounds. Ceramic processing also often involves a glass phase, which may be a desirable or undesirable product, while refractories are limited by melting. In all these cases, phase equilibria and reactions are closely linked to the acid–base properties of the melts and of the oxides which dissolve in them. It is therefore fruitful to ask whether the systematics developed above for acid–base chemistry in simple low-melting oxide solvents follow analogous trends when the solvent is a multicomponent aluminosilicate melt at a higher temperature.

There have been a number of calorimetric studies of the enthalpies of solution (mainly drop solution experiments) of oxides in aluminosilicate melts typically using commercially available calorimeters of the modified Calvet-type such as the Setaram MHTC96 instrument. We present the selected data below.

Figure 7 contains the enthalpies of solution (per mole of oxide ion) of silica vs. the mole fraction of silica (or alumina plus silica) for a variety of melts [46,55]. The enthalpy of solution becomes more exothermic with decreasing acidity, reflecting a decrease in the thermodynamic activity of silica as its concentration decreases. Although the data show a scatter for different melt compositions, the overall trend is clear.

Motivated by the importance of the corrosion of jet engine surfaces by heated silicate dusts, Costa et al. [47] determined the enthalpies of solution of a large number of oxides in CMAS melts of several compositions. Figure 8a shows the enthalpies of solution of some selected oxides in CMAS systems differing in the content of the acid component (SiO_2_) by about seven values in mole %. With an increase in electronegativity, the difference in the enthalpies of solution of oxides (Figure 8b) decreases and practically disappears. Calcia, the most basic oxide, has the largest difference, and it dissolves more exothermically in a more acidic melt. For magnesia, this difference is already smaller, although its enthalpy of solution is also more exothermic in the more acidic melt. There are practically no differences in the enthalpies of solution in these two melts for silica.

The solution calorimetry of an oxide in a silicate melt represents an acid–base reaction, and so does the formation of a ternary oxide phase from its binary oxide components. In the latter case, the bigger the difference in acid–base character between the components, the more energetically stable the ternary compound. Thus, for example, the energetic stability of alkali silicates is greater than that of alkaline earth silicates and that of both are greater than that of rare earth silicates (on a comparable molar basis., e.g., one mole of oxide ions) [10].

## 8. Conclusions

Acid–base concepts are very useful for systematizing the energetics of oxides. An extensive set of data on enthalpies of solution of binary oxides in molten lead borate, sodium molybdate, and other molten oxides used as calorimetric solvents permits the identification of trends of energetics versus measures of acid–base character, including electronegativity. These ideas are extended to multicomponent silicate melts.

## Figures and Tables

**Figure 1 molecules-28-04623-f001:**
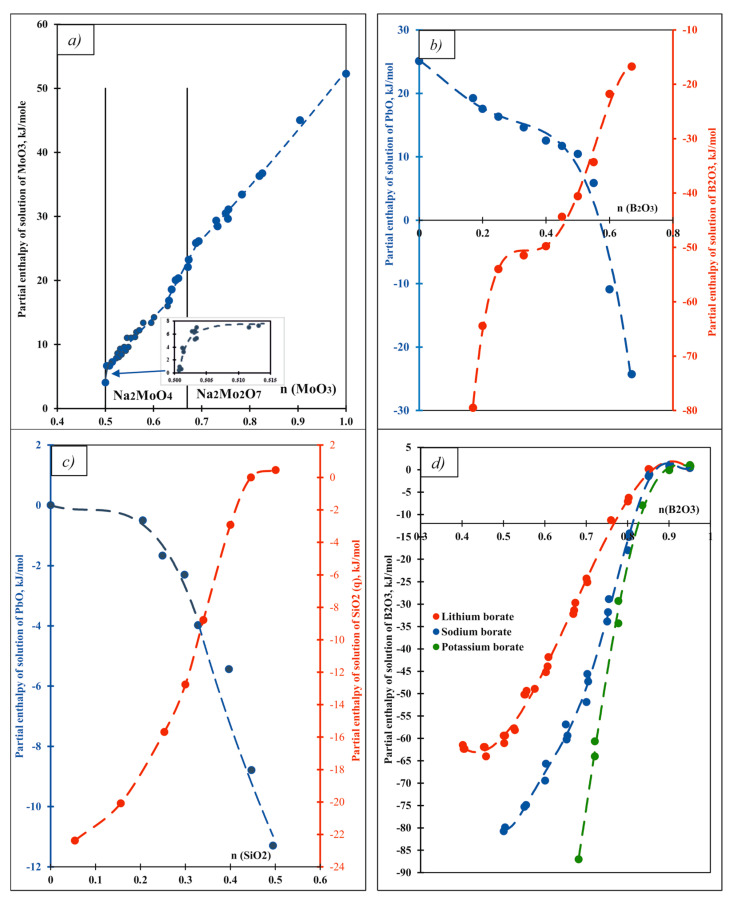
Partial molar enthalpies of solution as a function of the mole fraction of (**a**) MoO_3_ in Na_2_MoO_4_–MoO_3_ mixtures (melts) at 700 °C [45]; (**b**) liquid B_2_O_3_ and solid PbO in lead borate melts at 800 °C [42]; (**c**) liquid PbO and solid SiO_2_ (quartz) in lead–silicate melts at 900 °C [43]; (**d**) B_2_O_3_ in Li, Na, K borate melts at 667, 712 and 712 °C, respectively [27].

**Figure 2 molecules-28-04623-f002:**
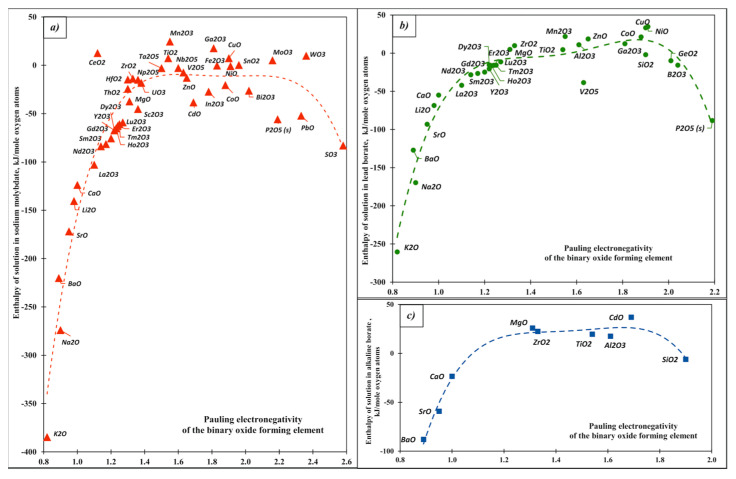
Enthalpies of solution of binary oxides vs. Pauling electronegativity: (**a**) trend for sodium molybdate melt at 700°C. (**b**) trend for lead borate at 800 °C. (**c**) The curves guide the eye.

**Figure 3 molecules-28-04623-f003:**
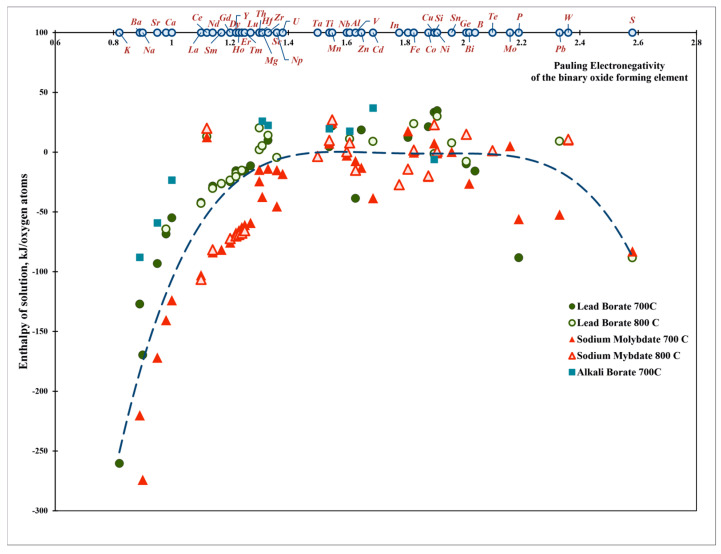
Enthalpies of solution of the binary oxides vs. Pauling electronegativity for different calorimetric solvents.

**Figure 4 molecules-28-04623-f004:**
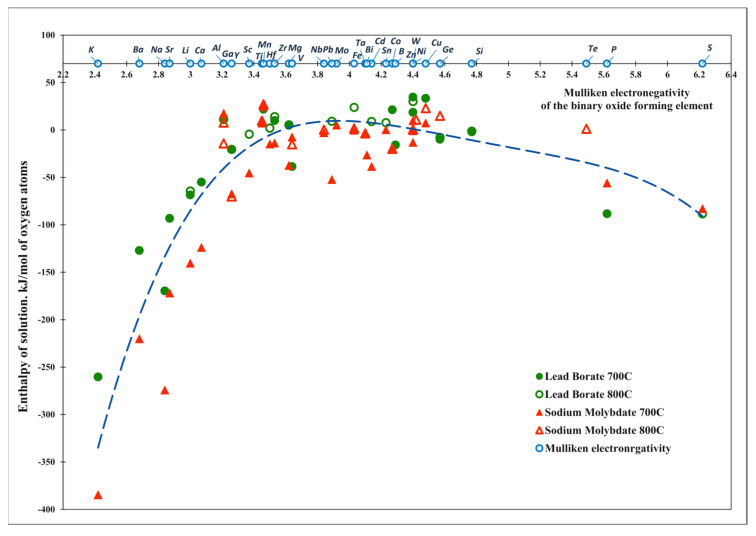
Enthalpies of solution of the binary oxides vs. Mulliken electronegativity for different calorimetric solvents.

**Figure 5 molecules-28-04623-f005:**
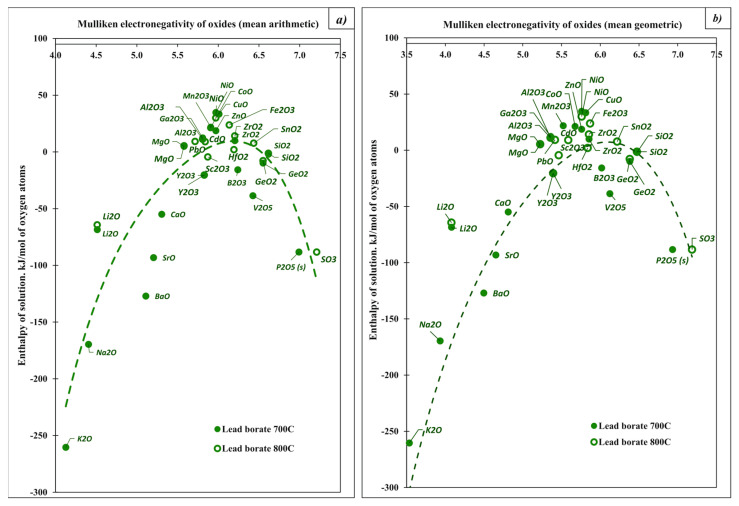
Enthalpies of solution of binary oxides in lead borate vs. Mulliken electronegativity [51] of oxides, calculated as arithmetic mean (**a**) and geometric mean (**b**) of electronegativity of consistent elements.

**Figure 6 molecules-28-04623-f006:**
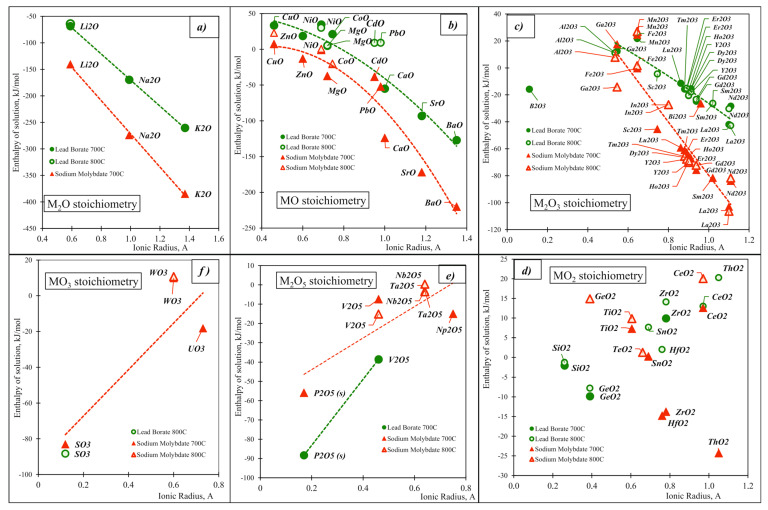
Enthalpies of solution of oxides, organized in groups according to stoichiometry of highest oxidation state of oxide-forming elements in lead borate and sodium molybdate melts vs. ionic radii of the oxide forming elements: (**a**) M_2_O stoichiometry, (**b**) MO stoichiometry, (**c**) M_2_O_3_ stoichiometry, (**d**) MO_2_ stoichiometry, (**e**) M_2_O_5_ stoichiometry, and (**f**) MO_3_ stoichiometry.

**Figure 7 molecules-28-04623-f007:**
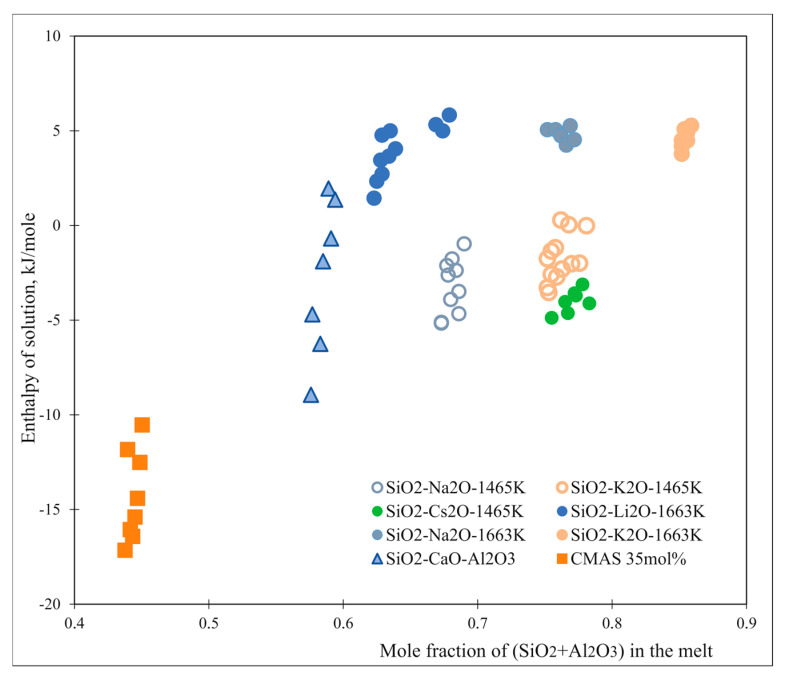
Enthalpies of solution of SiO_2_ in silicate melts vs. mole fraction of (silica plus alumina) in the silicate-containing melts [46,55].

**Figure 8 molecules-28-04623-f008:**
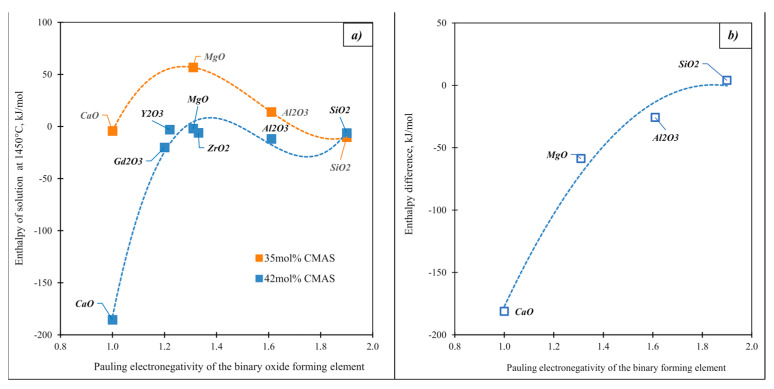
Enthalpies of solution (kJ/mole oxygen atoms) of the binary oxides (**a**) and enthalpy difference of enthalpy of solution of the binary oxides (**b**) in 35 and 42 mol.% silica in the CMAS systems.

**Table 1 molecules-28-04623-t001:** Heat contents (H^0^(T)-H^0^(25 °C), kJ/mol), enthalpies of drop solution ($\Delta_{ds}H$, kJ/mol) and enthalpies of solution ($\Delta_{s}H$, kJ/mole of oxygen atoms) of the binary oxides in lead borate and sodium molybdate at 700 and 800 °C.

Oxide	Pauling Electro-Negativity ^α^	Heat Content H^0^(T)-H^0^(25 °C), kJ/mol		Enthalpy of Drop Solution (kJ/mol) and Enthalpy of Solution (kJ/mol of Oxygen Atoms)			
				Lead Borate (2PbO·B_2_O_3_)		Sodium Molybdate (3Na_2_O·4MoO_3_)	
		700 °C	800 °C	$\frac{\Delta_{ds}H{\left(700℃\right)}^{\beta }}{\Delta_{ds}H{\left(800℃\right)}^{\beta }}$	$\frac{\Delta_{s}H{\left(700℃\right)}^{\gamma }}{\Delta_{s}H{\left(800℃\right)}^{\gamma }}$	$\frac{\Delta_{ds}H{\left(700℃\right)}^{\beta }}{\Delta_{ds}H{\left(800℃\right)}^{\beta }}$	$\frac{\Delta_{s}H{\left(700℃\right)}^{\gamma }}{\Delta_{s}H{\left(800℃\right)}^{\gamma }}$
K_2_O (*s*)	0.82	66.70 [28]	78.17 [28]	−193.68 ± 1.10 [11]	−260.38	−318.0 ± 3.1 [11]	−384.7

BaO (*s*)	0.89	35.55 [28]	41.20 [28]	−91.5 ± 1.9 [11]	−127.05	−184.61 ± 3.21 [11]	−220.16

Na_2_O (*s*)	0.9	56.52 [28]	68.76 [28]	−113.10 ± 0.83 [11]	−169.62	−217.56 ± 4.25 [11]	−274.08

SrO (*s*)	0.95	34.67 [28]	40.21 [28]	−58.5 ± 2.0 [11]	−93.17	−137.2 ± 3.8 [11]	−171.87

Li_2_O (*s*)	0.98	50.20 [28]	58.52 [28]	−18.28 ± 2.17 [11]	−68.48	−90.3 ± 2.5 [11]	−140.5
				−5.85 ± 0.65 [29]	−64.37		
CaO (*s*)	1.00	33.55 [28]	38.87 [28]	−21.4 ± 1.9 [11]	−54.95	−90.3 ± 1.8 [11]	−123.85

La_2_O_3_ (*s*, *A*-*type*)	1.10	83.73 [30]	97.43 [30]	−42.3 ± 4.4 [11]	−42.01	−225.1 ± 3.16 [11]	−102.94
				−30.91 ± 0.61 [11]	−42.78	−221.81 ± 2.25 [11]	−106.41
CeO_2_ (*s*, *cub*)	1.12	49.19 [31]	57.33 [31]			74.37 ± 0.75 [11]	12.59
				83.34 ± 1.86 [11]	13.00	97.40 ± 0.98 [32]	20.04
Nd_2_O_3_ (*s*, *A*-*type*, *hex*)	1.14	88.09 [30]	102.78 [30]	3.04 ± 3.70 [11]	−28.35	−163.36 ± 3.44 [11]	−83.82
				11.82 ± 0.99 [11]	−30.32	−142.20 ± 0.83 [11]	−81.66
Sm_2_O_3_ (*s*, *B*-*type*, *mon.*)	1.17	91.15 [30]	106.17 [30]	11.5 ± 4.1 [11]	−26.55	−153.62 ± 2.86 [11]	−81.59
				27.3 ± 0.6 [11]	−26.29		
Gd_2_O_3_ (*s*, *C*-*type*, *cubic*)	1.20	82.58 [30]	95.86 [30]	8.4 ± 3.4 [11]	−24.73	−144.34 ± 1.35 [11]	−75.64
				25.7 ± 0.2 [11]	−23.39	−121.0 ± 3.2 [11]	−72.29
Dy_2_O_3_ (*s*, *C*-*type*, *cub*)	1.22	86.70 [30]	100.39 [30]	40.2 ± 1.2 [11]	−15.5	−114.88 ± 2.22 [11]	−67.19
				46.9 ± 0.4 [11]	−17.83		
Y_2_O_3_ (*s*, *C*-*type*, *cub*)	1.22	81.19 [30]	94.11 [30]	19.6 ± 1.1 [11]	−20.53	−120.74 ± 0.94 [11]	−67.31
				32.8 ± 0.8 [11]	−20.44	−116.3 ± 1.2 [11]	−70.14
Ho_2_O_3_ (*s*, *C*-*type*, *cub*)	1.23	84.13 [30]	97.35 [30]	35.1 ± 5.6 [11]	−16.34	−111.72 ± 3.68 [11]	−65.28
						−109.51 ± 1.84 [11]	−68.95
Er_2_O_3_ (*s*, *C*-*type*, *cub*)	1.24	82.91 [30]	96.13 [30]	35.3 ± 1.7 [11]	−15.87	−105.26 ± 2.48 [11]	−62.72
				50.6 ± 0.4 [11]	−15.18	−107.2 ± 1.8 [11]	−67.78
Tm_2_O_3_ (*s*)	1.25	85.71 [30]	99.06 [30]	38.6 ± 2.8 [11]	−15.70	−97.12 ± 2.38 [11]	−60.94
						−97.97 ± 1.10 [11]	−65.68
Lu_2_O_3_ (*s*, *C*-*type*, *cub*)	1.27	80.40 [30]	93.61 [30]	46.2 ± 1.2 [11]	−11.4	−96.90 ± 1.90 [11]	−59.10

ThO_2_ (*s*)	1.3	49.50 [33]	57.5 ± 1.7 [34]			0.89 ± 0.48 [31]	−24.31
				98.1 ± 1.7 [34]	20.3 ± 2.4		
HfO_2_ (*s*, *monocl*)	1.3	49.56 [30]	57.78 [30]			20.0 ± 2.2 [35]	−14.78
				61.75 ± 1.38 [11]	1.99		
MgO (*s*)	1.31	31.60 [28]	36.71 [28]	36.48 ± 0.50 [11]	4.88	−5.79 ± 0.15 [11]	−37.39
				42.09 ± 0.41 [11]	5.38		
ZrO_2_ (*s*)	1.33	47.07 [28]	54.68 [28]	66.93 ± 0.92 [11]	9.93	19.5 ± 0.9 [11]	−13.79
				82.9 ± 0.7 [11]	14.11		
Sc_2_O_3_ (*s*)	1.36	78.25 [30]	91.21 [30]			−57.72 ± 0.98 [11]	−45.32
				77.83 ± 1.89 [11]	−4.46		
Np_2_O_5_ (*s*)	1.36	109.26 [33]				34.22 ± 5.34 [33]	−15.00

γ-UO_3_ (*s*)	1.38	64.22 [33]				9.49 ± 1.53 [36]	−18.24
				26.67 ± 4.02 [36]			
Ta_2_O_5_ (*s*)	1.5	110.71 [28]	129.29 [28]			95.8 ± 3.6 [37]	−2.98
						111.41 ± 1.61 [32]	−3.58
TiO_2_ (*s*, *rutile*)	1.54	46.26 [28]	53.70 [28]	55.4 ± 1.2 [38]	4.57	60.81 ± 0.11 [11]	7.28
						73.37 ± 0.36 [32]	9.84
Mn_2_O_3_ (*s*, *cub*)	1.55	81.10 [31]	94.75 [31]	146.60 ± 1.6 [11]	21.83	154.70 ± 1.00 [11]	24.53
						175.79 ± 1.38 [32]	27.01
Nb_2_O_5_ (*s*)	1.6	108.00 [28]	125.40 [28]			93.97 ± 1.60 [11]	−2.81
						127.05 ± 0.86 [32]	0.33
*α*-Al_2_O_3_ (*s*, *cor.*)	1.61	74.26 [28]	86.68 [28]	107.38 ± 0.15 [11]	11.04		
				120.12 ± 0.17 [11]	11.15	110.08 ± 1.17 [32]	7.80
V_2_O_5_ (*s*)	1.63	177.19(*l*) [28]	196.25(*l*) [28]	−15.92 ± 0.45 [11]	−38.62	140.0 ± 2.1 [11]	−7.44
						120.46 ± 0.57 [32]	−15.16
ZnO (*s*, *hex.*)	1.65	32.40 [31]	37.66	51.03 ± 0.36 [11]	18.63	19.4 ± 0.7 [11]	−13.00

CdO (*s*)	1.69	33.64 [39]	39.00 [39]			−4.82 ± 0.28 [11]	−38.46
				47.98 ± 0.81 [11]	8.98		
In_2_O_3_ (*s*)	1.78	81.11 [39]	93.98 [39]			−1.12 ± 0.25 [11]	−27.41
						12.23 ± 1.03 [11]	−27.25
Ga_2_O_3_ (*s*)	1.81	77.45 [30]	90.11 [30]	114.38 ± 1.17 [11]	12.31	130.16 ± 1.66 [11]	17.57
						132.46 ± 1.88 [11]	14.17
Fe_2_O_3_ (*s*)	1.83	96.43 (*II*-*modif*) [28]	110.94 (*III*-*modif*) [28]			95.63 ± 0.50 [11]	−0.27
				182.29 ± 1.34 [11]	23.78	115.92 ± 1.57 [32]	1.66
CoO (*s*)	1.88	36.25 [28]	41.80 [28]	57.48 ± 0.93 [11]	21.23	15.66 ± 0.59 [11]	−20.59
						21.92 ± 0.36 [32]	−19.88
CuO (*s*)	1.90	34.55 [28]	40.07 [28]	67.9 ± 0.6 [11]	33.35	41.9 ± 0.6 [11]	7.35
						63.05 ± 0.40 [32]	22.98
SiO_2_ (*s*, *quartz*)	1.90	43.46 [28]	50.37 [28]	39.4 ± 0.4 [11]	−2.03		
				47.79 ± 0.32 [11]	−1.29		
NiO (*s*)	1.91	36.60 (*cubic*) [31]	42.10 (*cubic*) [31]	71.3 ± 0.8 [11]	34.7	35.73 ± 0.95 [11]	−0.87
				72.08 ± 0.57 [11]	29.98	42.77 ± 0.35 [32]	0.67
SnO_2_ (*s*)	1.96	49.52 (*tetragon*) [31]	57.78 (*tetragon*) [31]			50.05 ± 0.21 [11]	0.27
				73.07 ± 0.86 [11]	7.65		
GeO_2_ (*s*, *quartz*)	2.01	44.76 [30]	52.29 [30]	25.06 ± 0.29 [11]	−9.85		
				36.7 ± 0.3 [11]	−7.80	82.11 ± 0.86 [11]	14.91
Bi_2_O_3_ (*s*)	2.02	84.23 (*s*) [31]	96.89 (*s*) [31]			5.21 ± 0.53 [11]	−26.34

B_2_O_3_	2.04	94.06 (*l*) [28]	107.03(*l*) [28]	46.75 ± 1.52 [29]	−15.77		

TeO_2_	2.1	48.84 (*s*) [39]	88.37 (*l*) [39]				
						90.89 ± 1.67 [40]	1.26
MoO_3_ (*s*)	2.16	60.96 (*s*) [28]	71.56 (*s*) [28]			76.21± 1.47 [11]	5.08

P_2_O_5_ (*s*)	2.19	115.00 [28]		−326.48 ± 1.21 [11]	−88.30	−164.60 ± 0.85 [11]	−55.92

PbO (*red*, *tetragon*)	2.33	36.86 [28]	42.88 [28]			−15.39 ± 1.14 [11]	−52.25
				52.07 ± 0.43 [11]	9.19		
WO_3_ (*s*)	2.36	62.15 (*orthorhomb*) [28]	73.61 (*tetragon*) [28]			91.68 ± 1.34 [11]	9.84
						105.78 ± 0.87 [32]	10.73
SO_3_(*g*)	2.58	45.43 [28]	53.11 [28]			−203.7 ± 4.091 [11]	−83.04
				−211.95 ± 3.55 [11]	−88.35		

***^α^***—Pauling electronegativity of the element which form the binary oxide [1]. ***^β^***—number of decimal places reduced to one for consistency between different reports. ***^γ^***—data per mole of oxygen atoms in the binary oxide.

**Table 2 molecules-28-04623-t002:** Enthalpies of solution of oxides (per mole of oxygen atoms) in lithium–sodium borate at 800 °C and in calcia–magnesia–alumina–silica (CMAS) systems at 1450 °C.

Oxide	Electro-Negativity (Pauling) [1]	Enthalpy of Solution in Lithium–Sodium Borate (LiBO_2_–NaBO_2_ Eutectic) (800 °C)	Enthalpy of Solution in CMAS with 35 mol.% Silica (1450 °C) [46]	Enthalpy of Solution in CMAS with 42 mol.% Silica (1450°C) [47]
BaO	0.89	−87.94 [48]		
SrO	0.95	−59.15 [48]		
CaO	1	−23.39 [48]	−4.30	−185.48
Gd_2_O_3_	1.2			−20.12
Y_2_O_3_	1.22			−3.03
MgO	1.31	25.94 [49]	56.70	−2.07
ZrO_2_	1.33	22.47 [48]		−6.07
TiO_2_	1.54	19.595 [48]		
Al_2_O_3_	1.61	17.53 [50]	13.87	
CdO	1.69	36.99 [48]		
SiO_2_	1.90	−6.065 [49]	−10.20	

Heat content of corresponding oxides for calculations of enthalpies of solution from drop solution enthalpies taken from Table 1.

## Data Availability

Not applicable.

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
