# Peer review of "Acid–Base Properties of Oxides Derived from Oxide Melt Solution Calorimetry"

_molecules, 2023, doi:10.3390/molecules28124623_

Round 1
Reviewer 1 Report
Authors provided a very robust and thorough work analyzing relationships of acid-base interactions in various systems with their thermodynamics. Obviously the updated set of heat of solution values is obviously will be very useful for scientists performing oxide melt solution calorimetry. The paper is recommended for publication, there are only some minor technical comments.
Page 2 line 77. There are extra space in the word “Substances”
Page 9 Figure 1 is covering part of the text
Figures 2 and 3: It’s understandable that there are lots of labels, but some of them are overlapping each other and the points making them almost unreadable.
References: Different scripts for “years” are used in the list of references.
Author Response
Reviewer 1
Authors provided a very robust and thorough work analyzing relationships of acid-base interactions in various systems with their thermodynamics. Obviously, the updated set of heat of solution values is obviously will be very useful for scientists performing oxide melt solution calorimetry. The paper is recommended for publication, there are only some minor technical comments.
Thank you
Page2 line 77. There are extra space in the word “Substances”
Removed
Page 9 Figure 1 is covering part of text
Corrected
Figure 2 and 3: It’s understandable that there are lots of labels, but some of them are overlapping each other and the points making them almost unreadable.
We have corrected them as much as possible
References: Different scripts for “years” are used in the list of references
Bold script was used only for journals and not for books.
Reviewer 2 Report
Dear Authors,
The manuscript under consideration is a bright review on thermodynamic properties of oxide systems studied by the high temperature solution calorimetric method. The study under review was carried out by the best present-day scientific thermodynamic school headed by Professor Alexandra Navrotsky. The main feature of experimental data on high temperature behavior of a wide spectrum of oxide systems presented in the reviewed manuscript is the highest level of the reliability and quality of the thermodynamic data discussed. It should be underlined that any kind of thermodynamic data such as enthalpy of solutions of binary oxide systems at high temperatures summarized in the manuscript may be a valuable and necessary addition to all modern thermodynamic data bases used at present for modeling of phase diagrams in the frame of CALPHAD approach.
The most fruitful theory for the discussion of the results presented in the manuscript used soundly and reasonably by the Authors is the acid-base theory originally suggested for consideration of oxide systems by Academician D.S. Korzhinsky. However, analyzing the list of references dealing with the problem of the acid-base interactions in oxide systems mentioned in the manuscript considered it will be reasonable probably in the Reviewer’s opinion to mention also the studies that were carried out in the scientific school headed by Academician M.M. Shultz and his colleagues. The most valuable among them may be for example the following.
1. Shultz M.M., Vedishcheva N.M., Shakhmatkin B.A. Acid-base concept in oxide glasses. Thermochimica Acta. 1987. V. 110. P. 443-447.
2. Shultz M.M. Acid-base concept as applied to oxide melts and glasses and the Mendeleev theory of the vitreous state. Fizika I Khimiya Stekla. 1984. V. 10. N 2. P. 129.
Knowledge of acid-base interactions in oxide systems was successfully used also for the developing of the acid-base concept of vaporization of oxide systems summarized partially in the following reviews, in which the decisive key of the differences of electronegativities, the bond energies evaluated when the second coordination sphere was taken into consideration as well as some other factors were discussed.
1. Stolyarova V.L. High temperature mass spectrometric study of oxide systems and materials. Rapid Communications in Mass Spectrometry. 1993. V. 7. N 11. P. 1022-1032. DOI: 10.1002/rcm.1290071112
2. Stolyarova V.L., Semenov G.A. Mass spectrometric study of vaporization processes of oxide systems. Ed. J.H. Beynon, F.R.S. Wiley & Sons, Ltd, Chichester. 1994. 434 p.:
Chapter “Vaporization processes and Acid-Base Interactions in Oxide Melts”, P. 320-333.
According to the Reviewer’s opinion this remark in no way lessens the highest quality of the presentation of the manuscript content but probably will be useful to the Authors in the further studies.
Thus, the manuscript by A. Navrotsky and A. Koryttseva completely meets the requirements for publication in the MDPI journal “Molecules”.
Author Response
Reviewer 2
The manuscript under consideration is a bright review on thermodynamic properties of oxide systems studied by the high temperature solution calorimetric method. The study under review was carried out by the best present-day scientific thermodynamic school headed by Professor Alexandra Navrotsky. The main feature of experimental data on high temperature behavior of a wide spectrum of oxide systems presented in the reviewed manuscript is the highest level of reliability and quality of the thermodynamic data discussed. It should be underlined that any kind of thermodynamic data such as enthalpy of solutions of binary oxide systems at high temperatures summarized in the manuscript may be a valuable and necessary addition to all modern thermodynamic data bases used at present for modeling of phase diagrams in the frame of CALPHAD approach.
The most fruitful theory for the discussion of the results presented in the manuscript used soundly and reasonably by the Authors is the acid-base theory originally suggested for consideration of oxide systems by Academician D.S.Korzhinsky. However, analyzing the list of references dealing with the problem of the acid-base interactions in oxide systems mentioned in the manuscript considered it will be reasonable probably in the Reviewers opinion to mention also the studies that were carried out in the scientific school headed by Academician M.M.Shultz and his colleagues. The most valuable among them may be for example the following.
- Shultz M.M., Vedishcheva N.M., Shakhmatkin B.A. Acid-base concepts in oxide glasses. Thermochimica Acta.1987.V.110.P.443-447.
- Shultz M.M. Acid-base concept as applied to oxide melts and glasses and the Mendeleev theory of the vitreous state. Fizika I Khimiya Stekla. 1984. V.10. N2. P.129.
Knowledge of acid-base interactions in oxide systems was successfully used also for the developing of the acid-base concept of vaporization of oxide systems summarized partially in the following reviews, in which the decisive key of the differences of electronegativities, the bond energies evaluated when the second coordination sphere was taken into consideration as well as some other factors were discussed.
- Stolyarova V.L., High temperature mass spectrometric study of oxide systems and materials. Rapid communications in Mass Spectrometry.1993.V.7. N. 11.P. 1022-1032. DOI: 10.1002/rcm/1290071112.
- Stolyarova V.L., Semenov G.A. Mass spectrometric study of vaporization processes of oxide systems. Ed. J.H Beynon, F.R.S. Wiley & Sons, Ltd, Chichester. 1994. 434 p.: Chapter “Vaporization processes and Acid-Base Interactions in Oxide Melts”, P. 320-333.
According to Reviewers opinion, this remark in no way lessens the highest quality of the presentation of the manuscript content, but probably will be useful to the Authors in the further studies.
Thus, the manuscript by A. Navrotsky and A. Koryttseva completely meets the requirements for publication in the MDPI journal “Molecules”.
Additional references have been added.
Reviewer 3 Report
This study is devoted to the melt solution calorimetry results and also attempt to unite results from previous studies on the base of electronegativity conception. The experimental work reported in the manuscript already was published in previous article and discussion of the results is rather poor hence I recommend to reject article in the present form.
1. Introduction must be shortened. The most part of introduction is well known conceptions of various acid-base theories from common course of general chemistry. It more situated for a textbook but not for research article or even review. If authors want to mention these theories they may refer to any well-known textbook.
2. The main question to proposed conception is why authors choose Pauling electronegativity scale? There are many other electronegativity scales most known Mulliken and Allred–Rochow scales. I think authors must to discuss and substantiate their selection of Pauling scale. And also may to use another scales. Moreover there are some scales not for element only and even for oxides (see for example L.M. Viting, Vysokotemperturnye rastvory-rasplavy (Нigh temperature solutions-melts), Moscow University Publ., Moscow, 1991. in Russian).
3. Authors associate oxide electronegativity with electronegativity of corresponding element. How, in this case, can one distinguish between oxides of the same element in different oxidation states? For example, Ce2O3 and CeO2, FeO and Fe2O3 and so on.
4. In page 10 line 311 authors mention term “melt acidy”. What do they mean? And what is quantitative measure of the acidity of a melt?
5. Page 11 line 348 and above. Authors says that “the dependencies well described by a polynomial of the second degree and higher degree. And increase of the polynomic degree increases approximation confidence value”. This is simple feature of the polynomial approximation. If author take the polynomic degree equal to the number of experimental point the approximation would be ideal.
6. Page 11 line 351. So called “Õ shape” of approximation is definite by only one point for SO3. What is experimental error of the determination value of this point? Because if it would be removed the “Õ shape” disappear.
7. English must be improved through all manuscript. Some of sentences are interlinear translation of Russian terms. For example, “Heat content” function which is enthalpy increment. Some od sentences I do not understand: for example, what means “many close to zero” and “More limited data” (see abstract first page, line 21)
8. Some minor corrections:
Page 2 Line 46. What means “Stable high melting compounds”? It means that all stable compounds are high melting?
Page 2 Line 77. Remove space between parts of the word “Substances”
Page 2 Line 86. The formula of orthosilicate anion is SiO44-
Page 3 Line 109. Quotion mark before “generalized”
Page 5 line 183 Index “f” before D is needed in the formula
Page 6 Line 275. I think authors means Table 1 not figure 1
Table 1. Remove bracket in line 276. What means PbBO and NaMoO in the table title?
Some data on enthalpy increment (UO3 for example) may be taken from Thermodynamic Properties of Individual Substances V.P Glushko (Ed.), Nauka, Moscow, 1978−1982 handbook and not from original literature.
Page 9 line 282. This reaction was already mentioned in introduction.
Page 9 Line 288 Obviously third reaction is wrong. It does not have mass and charge balances.
Figure 1. What is on X axis? Mole fraction or what?
Page 13 line 373 change dots to commas after substances
Page 14 line 388 MO or alkaline earth? Because authors mention some oxides of MO composition such as ZnO, CdO which are not alkaline earth oxides.
Author Response
Reviewer 3
This study is devoted to the melt solution calorimetry results and also attempt to unite results from previous studies on the base of electronegativity conception. The experimental work reported in the manuscript already was published in previous article and discussion of the results is rather poor hence, I recommend to reject article in the present form.
This is intended to be a review article, which is encouraged by the journal.
Introduction must be shortened. The most part of introduction is well-known conceptions of various acid-base theories from common course of general chemistry. It more situated for a textbook but not for research article or even review. If authors want to mention these theories, they may refer to any well-known textbook.
When preparing the Introduction, not only books were used, but also original articles, since not all textbooks describe all acid-base theories. Furthermore, some of the older Russian references may not be familiar or accessible to the Western audience, so referring to them is useful. Nevertheless, at the reviewer’s suggestion, we have shortened this section somewhat.
The main question to proposed conception is why authors choose Pauling electronegativity scale? There are many other electronegativity scales most known Mulliken and Allred-Rochow scales. I think authors must to discuss and substantiate their selection of Pauling scale. And also may to use another scales. Moreover there are some scales not for element only and even for oxides (see for example L.M.Viting. Vysokotemperaturnye rastvory-rasplavy (High temperature solutions-melts), Moscow University Publ., Moscow, 1991, in Russian)
We have added a paragraph to the text explaining why we picked the scale to use, mentioning other ones, and explaining they give consistent correlations-. We have also added several scales to the figure to show that all show similar trends.
We put a new paragraph on page 10 after line 278: “When choosing the EN scale we decided to use the Pauling scale, because the electronegativities of almost all elements are given. Moreover, all EN scales are in good agreement with each other. For comparison, We plot heat of solution from the Mulliken EN of elements [51] (Figure 4), and obtain similar results as when using Pauling scale: both in terms of the shape of the curve and in polynomial fits.
The EN concept applies not only to elements, but also to ions and compounds, including oxides. There are different approaches to the calculation of EN of compounds: arithmetic mean [52] and geometric [53] from the consistent elements. Plots of enthalpy of solution vs. these EN parameters give similar trends as those calculated using EN of elements (Figure 5 a-b).” Accordingly, new Figure 4 and Figure 5 were plotted and added to the paper.
Authors associate oxide electronegativity with electronegativity of corresponding element. How in this case, can one distinguish between oxides of the same element in different oxidation states? F.e., CeO2 and Ce2O3, FeO and Fe2O3 and so on.
All oxides in the paper and on the graphs are only in their highest oxidation states. The remaining oxides were excluded from the consideration. This is due to the fact that oxides in lower oxidation states will be oxidized to higher oxidation states in sodium molybdate and lead borate melts, making an additional exothermic contribution to the enthalpy of solution in the form of the enthalpy of oxidation.
In page 10, line 311authors mention term, “melt acidy”. What do they mean? And what is quantitative measure of the acidity of a melt?
This is a misprint that has been corrected. This is definitely “melt acidity”. We assume mole % of silica or silica plus alumina in the melt to be a good representation of the acidity of the melt.
Page 11, line 351, so called “Π-shape” is definite by only one point for SO3. What is experimental error of the determination value of this point? Because if it would be removed the “Π-shape” disappear.
The experimental error in determining the enthalpy of solution of SO3 is the same as for the others, 1-2%.
English must be improved through all the manuscript. Some of sentences are interlinear translation of Russian terms. For example “heat content” function which is enthalpy increment. Some od sentences I do not understand: for example, what means “many close to zero” and More limited data (see Abstract first page, line 21).
The term heat content is routinely used in the American literature. We have gone through the paper and cleaned up the writing in general.
Some minor corrections
Page 2 line 46. What means “Stable high melting compounds”? It means that all stable compounds are high melting?
High melting compounds are generally stable.
Page 2 line 77. Remove space between parts of the word “Substances”
Corrected
Page 2 line 86. The formula of orthosilicate anion is SiO44-
Corrected
Page 3 line 109. Quotation mark before “generalized”.
Added
Page 5 line 183.Index “f” before D is needed in the formula
Replaced “ox” by “f,ox” in lines 181, 183, 187, 188 and 191, everywhere the corresponding values present.
Page 6 line 275. I think authors means Table 1 not figure 1
No, we mean Figure 1, not Table 1.
Table 1. Remove bracket in line 276. What means PbBO and NaMoO in table title?
It means lead borate and sodium molybdate, but we have removed these abbreviations for clarity
Some data on enthalpy increment (UO3 for example) may be taken from Thermodynamic Properties of Individual Substances V.P.Glushko (Ed.), Nauka, Moscow, 1978-1982 handbook and not from original literature.
Glushko handbook gives enthalpy increment of UO3 for H(298K)-H(0K) and not for H(700C)-H(25C), which is needed in the paper, to derive enthalpy of solution from the enthalpy of drop solution.
Page 9 line 282. This reaction was already mentioned in Introduction.
There is no such reaction in the Introduction.
Page 9 line 288. Obviously third reaction is wrong. It does not have mass and charge balances.
Corrected to MoO3 (s) = MoO3 (in melt)
Figure 1. What is on X axis? Mole fraction or what?
Correct caption for mole fraction
Page 13 line 373 change dots to commas after substances
Corrected
Page 14 line 388 MO or alkaline-earth? Because authors mention some oxides of MO composition such as ZnO, CdO which are not alkaline-earth oxides.
MO means all oxides with such a stoichiometry, namely: alkaline-earth oxides together with d-block elements oxides. So, we added “and d-block element oxides” .
Other comment from authors
Concerning the English, a reviewer or staff member made the unwarranted and possibly prejudiced assumption that, because the authors’ names look foreign, they are not native English speakers. Navrotsky is in fact a native speaker (born in the USA). In the process of revision, we have checked and edited the manuscript for clarity and English, without changing its basic style. We trust this is satisfactory.
Round 2
Reviewer 3 Report
See attachment

Author Response
Navrotsky an Koryttseva response to reviewer 3 5.31.23
This study is devoted to the melt solution calorimetry results and also attempt to unite results from previous studies on the base of electronegativity conception. Authors improved the paper but there are still some questions yet. I recommend to publish article after minor revision.
- Page 3 Line 101. “Therefore, the strength of a base should decrease with its EN… whereas the strength of an acid should increase…”. Rather than expand this sentence, we have eliminated it as the main thought is already in the prior sentence.
Generally speaking this is not true. Because we cannot divide affection of atom under consideration and other atoms in the compound. For example, the strength of the acid in HF-HCl-HBr-HI row is increase whereas the EN of halogen is decrease or in the BF3-BCl3-BBr3 row the weakest acid is BF3. I think this statement may be written with more exactness. Also, I suggest to add “increasing” after EN.
- Page 10 line 277. “High degree polynomials fit the data better as expected when one use more parameters”.
It’s nothing out of the ordinary. If to take polynomial order equal to the number of experimental points then ideal regression will be obtained. I think this statement have to removed. Removed and section slightly rewritten to read “Taken individually for specific systems (Figure 2) or together for all systems (Figure 3), the heat of solution data shows roughly parabolic trends with electronegativity. And the dependence is adequately described by a quadratic polynomial. We stress that this polynomial (or another one of higher order to better fit the data) has no physical significance; it is an aid to the eye and may offer a…..
- Concerning Figure 3 and 4. Why the numbers of the experimental points at figure 3 more than for figure 4? Is it because of Mulliken EN values (in particular atoms electron affinities) is not known for some elements? If so authors may take these values for example from this paper “Electron Affinities of Atoms and Structures of Atomic Negative Ions” https://doi.org/10.1063/5.0080243 Yes that is the reason there are fewer points. We are concerned that the parameters in the suggested reference may not be quite the same in terms of calculation as the Mulliken values, so we have chosen to not compare them. The figures make the needed points already.
- All oxides in the paper and on the graphs are only in their highest oxidation states. The remaining oxides were excluded from the consideration. This is due to the fact that oxides in lower oxidation states will be oxidized to higher oxidation states in sodium molybdate and lead borate melts, making an additional exothermic contribution to the enthalpy of solution in the form of the enthalpy of oxidation
Not fully correct statement. For example, PbO, MnO2, CoO, TeO2 are not highest oxides. Of course, most of them thermally unstable at the experiment conditions. May be better to use term “oxides stable at the conditions of the experiment”? changed as suggested.
- Glushko handbook gives enthalpy increment of UO3 for H(298K)-H(0K) and not for H(700C)-H(25C), which is needed in the paper, to derive enthalpy of solution from the enthalpy of drop solution.
Authors may recalculate data from this handbook by simple linear interpolation of given values. Thus, for UO3 from Glushko handbook data for 700 C (973 K) I obtain value 64.08 kJ for enthalpy increment which is in good agreement with value given by authors 64.22 kJ. For 800 C (1073 K) the value is 74.36 kJ. Trying to correct or extrapolate the data in the handbook seems a little arbitrary and not a main point and we have left this section as it is.
- High melting compounds are generally stable.
I think that melting point and thermal stability of a compound do not have any correlation. For example, carbon oxide is stable but low melting compound. I suggest to add “and” between “stable” and “high melting”. “and” is added.
Figure 1. What is on X axis? Mole fraction or what? Correct caption for mole fraction
Caption is corrected to “Partial molar enthalpies of solution as a function of mole fraction….”
We have also corrected a few more typos and made minor writing edits (not marked)
